# Effect of Drought and Rehydration on Physiological Characteristics of *Agriophyllum squarrosum* (L.) Moq. in Different Habitats

**DOI:** 10.3390/plants13182601

**Published:** 2024-09-18

**Authors:** Juanli Chen, Yongqing Luo, Xueyong Zhao, Yan Li, Junpeng Mu

**Affiliations:** 1Ecological Security and Protection Key Laboratory of Sichuan Province, Mianyang Normal University, Mianyang 621000, China; juanlic@163.com (J.C.); luoyongqing8401@sina.com (Y.L.); leeleehi@163.com (Y.L.); 2Northwest Institute of Eco-Environment and Resources, Chinese Academy of Sciences, Lanzhou 730000, China; zhaoxy@lzb.ac.cn

**Keywords:** *Agriophyllum squarrosum*, drought and rehydration, physiological responses, antioxidant activity, desert steppe and sandy land

## Abstract

*Agriophyllum squarrosum* (L.) Moq. is a highly prevalent xerophytic species found throughout northern China. It is suitable for cultivation in semi-arid sandy environments and may establish roots in arid desert locations. This species plays a pioneering and exploratory role in the colonization of desert plants. In this study, we selected *A. squarrosum* from the Urat desert steppe (UD) and Horqin sandy land (HS) to explore their adaptation mechanisms to drought and rehydration environments by using the pot weighing control method to simulate an arid environment. The findings showed that the control (watering to 60–65% of field capacity) exceeded its required amount and the leaves turned yellow. The chlorophyll content was lower than those under moderate and severe drought, and rehydration caused a decrease. However, the contents of malondialdehyde, soluble sugar, and proline in the drought treatment were higher than those in the control. Under moderate and severe drought, the chlorophyll content and the quantum efficiency of photosystem II (*F*v/*F*m) of *A. squarrosum* from UD were higher than those from HS. During drought and rehydration processes, the proline content was relatively lower, while the activities of peroxidase (POD), superoxide dismutase (SOD), and catalase (CAT) and the content of soluble sugar and soluble protein were higher. However, antioxidant enzymes and osmoregulators from UD were higher than those from HS. The results suggest that the stronger ability of *A. squarrosum* to endure drought environments in UD is due to the high level of antioxidant enzymes and osmoregulators, which are conducive to relieving cell membrane damage when subjected to drought and rehydration.

## 1. Introduction

The global arid and semi-arid areas occupy one-third of the terrestrial land, and half of China is made up of arid and semi-arid regions [1]. Water is the main limiting factor for productivity development and ecological restoration in these areas, and the resulting drought stress affects plant growth and development, along with physiological and biochemical metabolic processes [2]. Extreme drought and intense precipitation events often arise as a result of fluctuations in global temperature and precipitation patterns, and there is even a phenomenon known as alternating drought and flood [3,4]. Extreme drought results in increasingly high water scarcity for plants, which has a serious impact on the structure, function, and biodiversity of ecosystems [5]. When plants face drought stress, their ability to grow normally hinges not only on their inherent drought resistance but also on their ability to recover from the drought and rehydrate [6]. The ability to recover after a drought is also an important parameter for evaluating their adaptability to drought environments. Whether plants can adapt to climate variation is critical for understanding the effects of climate change on plant population dynamics [7].

Under drought conditions, plant absorption of water and nutrients is inhibited, and they can resist the short-term or long-term effects of drought stress by reducing the photosynthetic rate, slowing down growth rate and other physiological regulatory processes [8]. Plants in arid and semi-arid areas often suffer from drought stress in stages, alternating between drought and precipitation. Rehydration after drought has a restorative effect on plant growth and development, but many plants exhibit different recovery abilities after rehydration [9]. Research has shown that plants that have experienced drought and rehydration have higher stress resistance, water use efficiency, and a faster growth rate than those that have not been affected by drought. Previous studies have found that rehydration after moderate drought can restore growth to a certain extent, and some plants even exhibit overcompensation phenomena [10]. It is used in actual production, such as water control by squatting seedlings, and different types of limited irrigation. The antioxidant enzyme activity and osmoregulator content of *Setaria viridis* and *Digitaria sanguinalis* increased as the stress level worsened after repeated artificial drought, but rehydration generated a decrease [11]. Rehydration also reduced the MDA content and cell membrane permeability of *Artemisia halodendron* and *Pennisetum centrasiaticum*, restoring their physiological functions [12]. However, other studies have found that rehydration after drought does not alleviate the impact of drought on plants, and it still inhibits their growth and development [13]. Plants respond to habitat changes through diverse adaptive characteristics under adversity, and this plasticity can be manifested in morphology and physiology, which is also the key to plant survival in heterogeneous environments. It can be seen that the rapid growth and improved physiological antioxidant capacity of plants during drought are physiological plasticity responses to adapt to adversity and maintain sustainable survival [11,12].

*Agriophyllum squarrosum* (L.) Moq. is an annual herbaceous plant of the Amaranthaceae family with a wide range of ecological adaptability, and it is common in the barren soil of deserts and sandy areas in northern China that have an annual rainfall of 50–400 mm [14]. It has evolved multiple specific adaptive traits, such as thicker leaf cuticles, spiky tips at leaf tips, and dense trichrome throughout the body, which can reduce water evaporation [15]. *Agriophyllum squarrosum* invades quicksand habitats with its tolerance to sand burying and can also be successfully planted in mobile dunes with its rapid response to water environment. It enhances the physical and chemical characteristics of soil to promote the infiltration and establishment of other plants, earning it the title of a “sand fixation pioneer” [16]. The branches and leaves can still retain themselves on the surface after withering, and it still has good windproof and sand blocking effects in spring and winter when strong winds are prone to occur. It is the starting point of the desert ecosystem and plays an important role in the improvement and management of mobile dunes and the restoration and reconstruction of degraded sand vegetation [17]. In the context of global climate change, extreme weather occurs frequently [18], and there are certain differences in environmental adaptability of *A. squarrosum* in different habitats. Our study conducted a pot experiment to uncover the physiological response patterns of *A. squarrosum* from different habitats to drought and rehydration environments. The aim was to simulate the mechanisms of stress resistance and compensation in drought and rehydration conditions. These findings provide theoretical support for future ecological engineering and the construction of artificial vegetation.

## 2. Materials and Methods

### 2.1. Experimental Location and Plant Material

The Urat desert steppe (UD) is situated in the transitional zone between desert and typical steppe and experiences a continental arid climate, with an average annual precipitation of 180 mm [19]. The annual evaporation greatly exceeds the precipitation, often surpassing it by more than tenfold [20]. The primary vegetation consists of *Reaumuria songarica*, *A. squarrosum*, and *Salsola collina* [21]. The Horqin sandy land (HS) is located in the transition zone between agriculture and husbandry and experiences a continental semi-arid climate characterized by an average rainfall of 350 mm and an annual evaporation of 1935 mm. The primary floras are *A. squarrosum*, *Artemisia halodendron,* and *Salsola collina* [22,23]. Over the past two decades, the combination of land degradation and water constraints has posed challenges to vegetation restoration in both locations. In the contest of global warming, frequent extreme precipitation events have a significant impact on the community structure and function, along with the germination characteristics and physiological response of individuals in the UD and HS ecosystems.

Following maturation, the majority of *A. squarrosum* seeds do not separate from the parent plant and instead stay in the crown for an extended period until the coming of the windy season in April of the following year, which triggers their quick release [24]. In early January, fully grown branches of *A. squarrosum* were randomly selected from the Urat desert steppe and Horqin Sandy land, and transported to the laboratory. After manual threshing from the stalks, they were placed in a dry room with a temperature range of 15–25 °C for storage. Prior to sowing, each seed underwent a sterilization process using sodium hypochlorite (1% NaClO) for the duration of 40 min. Additionally, the seeds were cleaned four times with pure water [25].

### 2.2. Treatments and Experimental Design

In late April, the seeds were planted in plastic pots measuring 28 cm in width and 21 cm in depth. At first, we filled these pots with 9 kg of dried and sifted sandy soil. The planting took place under a shelter at the Naiman Desertification Research Station, which is part of the Chinese Academy of Sciences. We selectively reduced the number of seedlings after a 20-day germination period, retaining only 5 in each container. The remaining plants had similar development patterns and were cultivated in the same setting with adequate irrigation until the start of the experiment. The shelter’s light intensity fell by around 10% compared with the environmental level. However, due to its sufficient height of 4 m, both the internal and outdoor ambient temperatures were below 1 °C [12].

A total of 120 pots of *A. squarrosum* were subjected to three different drought gradients: a control group with a field capacity of 60–65%, a moderate drought group (MD) with a field capacity of 40–45%, and a severe drought group (SD) with a field capacity of 20–25%. We utilized the weighing method to regulate the daily soil moisture level and ensure it remained within the required range for a period of 10 days from 1 to 10 August. We randomly selected a total of 60 pots on 11, 17, and 23 August to investigate the impact of chronic drought stress on the physiological indicators of *A. squarrosum*. Each treatment was replicated 20 times. We rehydrated the remaining 60 pots on 17 and 23 August to assess the effects of watering after a drought period.

### 2.3. Analytical Methods and Statistical Analysis

The chlorophyll fluorescence parameters of *A. squarrosum* were initially determined by exposing dark-adapted leaf clips to a pulse-amplitude-modulation chlorophyll fluorometer (Hansatech, England, UK) for 25 min [26]. Subsequently, using scissors, we obtained a portion of the foliage situated above the ground level of the plants. A portion of the materials were promptly transported to the laboratory for the purpose of assessing the chlorophyll content. Subsequently, using a pair of scissors, we obtained a portion of the foliage situated above the ground level of the plants [23]. We extracted chlorophyll from recently harvested leaves using 80% acetone as the extracting agent. The concentration of chlorophyll was determined by measuring the absorbance values at wavelengths of 663 nm and 645 nm [27].

Blades fixed with liquid nitrogen were extracted using a chilled buffer (2% polyvinylpolypyrrolidone, 50 Mm phosphate) and the extract was centrifuged at 15,000× *g* for 20 min, and the supernatant was used for the determination of various physiological makers [1]. The concentration of malondialdehyde (MDA) was measured by spectrophotometry at wavelengths of 450 nm, 532 nm, and 600 nm, employing the thiobarbituric acid method [28]. The activities of antioxidant enzymes (POD, SOD, and CAT) were specifically quantified following the methodology described by An et al. [29]. POD activity was assessed using the guaiacol method, where the enzyme activity unit was defined as the oxidation of 1 μmol of guaiacol per gram of plant fresh weight per minute. SOD activity was assessed using the nitrogen blue tetrazole (NBT) photoreduction method, where 50% inhibition of NBT photoreduction per gram of plant fresh weight was considered as one unit of enzyme activity. CAT activity was assessed by hydrogen peroxidation iodimetry, with a rate of 1 μmol H_2_O_2_ decomposed per gram of fresh plant weight per minute serving as the unit of enzyme activity [11]. The osmoregulator contents were definitely measured in accordance with Luo et al. [30]. The concentration of soluble sugars was determined using the anthrone reaction. The concentration of soluble proteins was measured using the Coomassie brilliant blue method, whereas the concentration of proline was evaluated using the ninhydrin method [12]. The spectrophotometer (Shimadzu Corporation, Kyoto, Japan) was used to measure the activity of antioxidant enzymes, along with the contents of osmoregulatory substances and MDA.

The determination indices were reported as the mean ± standard error and examined using the analysis of variance (ANOVA) procedure and LSD Fisher’s test in SPSS release 20.0. We graphed the data using version 12.5 of SigmaPlot. We evaluated the correlations among physiological indices of *A. squarrosum* from various habitats using Origin Pro 2022.

## 3. Results

### 3.1. Changes in Chlorophyll, Fv/Fm, and Malondialhyde

The leaves of both species in the control turned chartreuse after natural drought (Table 1), and the chlorophyll content of two plants exhibited a positive correlation with the severity of stress (Figure 1A). During a natural drought, the chlorophyll content of *A. squarrosum* from UD was arranged in the following order: control, moderate drought, and severe drought. The content of it from UD was higher than that from HS under moderate and severe drought, but the control was opposite. With increasing drought duration, the chlorophyll content of *A. squarrosum* from UD increased initially and then decreased under moderate and severe drought, whereas the content in the control group decreased gradually. On the other hand, the chlorophyll content of *A. squarrosum* from HS decreased initially and then increased in the control group, while its content under moderate and severe drought decreased gradually. Both species saw a reduction in chlorophyll concentration as a result of rehydration. Furthermore, the chlorophyll content after 13 days of rehydration was lower than after 7 days of rehydration. On 24 August, the chlorophyll content of *A. squarrosum* from UD was 2.42 times higher under moderate drought and 2.97 times higher under severe drought compared with the control. In contrast, the chlorophyll content of *A. squarrosum* from the HS decreased by 27.72%, 17.88%, and 5.05% in the control, moderate, and severe drought conditions, respectively, compared with the content before rehydration.

As the stress level worsened and the duration lengthened, the *F*v/*F*m of the two species decreased, with *A. squarrosum* from the UD site showing a higher *F*v/*F*m than from the HS (Figure 1B). On 11 August, there was no significant difference in *F*v/*F*m among different treatments between the two plants. On 23 August, the *F*v/*F*m of *A. squarrosum* from UD (0.643) and HS (0.616) reached their lower values under severe drought, which decreased by 18.1% and 14.9% compared with on 11 August. Rehydration on 17 August caused an increase in *F*v/*F*m of the two species, which was higher than on 23 August. The *F*v/*F*m of *A. squarrosum* from UD in the control and moderate drought was lower than before rehydration on 17 August; the *F*v/*F*m of others were higher than before rehydration.

On 11 August, the malondialdehyde content of both species decreased as the stress level worsened (Figure 1C), and the content of *A. squarrosum* from HS under severe drought was 6.026 mmol·g^−1^ FW, which was significantly lower than the control. On the 7th and 13th days after drought, the malondialdehyde content of *A. squarrosum* from UD under moderate drought reached the maximum value (9.722 mmol·g^−1^ FW and 10.453 mmol·g^−1^ FW, respectively), which was significantly higher than the control and severe drought. The malondialdehyde content of *A. squarrosum* from HS under severe drought on 23 August was 7.521 mmol·g^−1^ FW, which was substantially more than the levels found in the control and moderate drought conditions. Furthermore, the rehydration process on 24 August resulted in a decrease in MDA content in *A. squarrosum* across various habitats.

### 3.2. Changes in Antioxidant Enzymes

On 11 August, the level of POD activity in *A. squarrosum* from UD did not show a significant difference between the control (58.129 H_2_O_2_·g^−1^ FW·min^−1^) and moderate drought (64.058 H_2_O_2_·g^−1^ FW·min^−1^). However, both species were significantly higher than the level seen during severe drought (Figure 2A). With increasing drought duration, the activity of both species initially decreased and then increased in control and moderate drought, while its activity increased continuously under moderate and severe drought. Rehydration resulted in an elevation of POD activity in *A. squarrosum* from various origins. On 18 August, the activity of *A. squarrosum* from HS in control, moderate, and severe drought increased by 15.07%, 10.92%, and 17.74%, respectively, compared with the activity before rehydration. Additionally, the activity after severe drought was significantly higher than in the control and moderate drought. On 24 August, POD activity of *A. squarrosum* from UD achieved its peak value of 113.768 H_2_O_2_·g^−1^ FW·min^−1^, which was 1.62 times and 1.20 times greater than those under moderate and severe drought, respectively.

SOD activity of the two species increased as the stress level worsened and the duration was prolonged (Figure 2B). During a natural drought, SOD activity of *A. squarrosum* from UD under severe drought was considerably higher than in the control and moderate drought. On 17 August, the activity of *A. squarrosum* from UD was 120.838 μg·FW^−1^·h^−1^. This value was 1.64 times higher than the activity in the control group and 1.32 times higher than the activity in moderate drought, and there was significant difference among the three treatments. Rehydration caused an elevation of SOD activity in *A. squarrosum* from UD, and its activity increased as the stress level worsened. SOD activity in *A. squarrosum* from HS was shown to decline by 8.06%, 17.23%, and 10.33% in the control, moderate, and severe drought on 18 August. However, the SOD activity increased by 16.02%, 9.03%, and 6.42% on 24 August under the same conditions.

CAT activity of *A. squarrosum* from UD was higher than from HS. The activity of *A. squarrosum* from UD showed an initial increase followed by a drop under moderate and severe drought as drought duration increased. In contrast, the control group exhibited the reverse pattern (Figure 2C). The activity of *A. squarrosum* from HS intensified as the stress level worsened. On 11 August, CAT activity of *A. squarrosum* from HS was the lowest, exhibiting a loss of 21.58% and 45.08% compared with moderate and severe drought. Furthermore, there were significant variations found among the treatments. Following rehydration, CAT activity from UD decreased, and the activity in moderate and severe drought was significantly higher than in the control. Rehydration generated a reduction in CAT activity of *A. squarrosum* from HS, and the activity in control, moderate, and severe drought was 1.02 times, 1.02 times, and 1.11 times higher than before rehydration on 24 August.

### 3.3. Changes in Osmoregulator

The content of soluble sugar of the two species increased as the stress level worsened and the duration was prolonged, and the content of *A. squarrosum* from UD was higher than from HS (Figure 3A). On 23 August, the content of soluble sugar in the categories of control, moderate, and severe drought was measured to be 49.968 μg·g^−1^ FW, 50.162 μg·g^−1^ FW, and 42.133 μg·g^−1^ FW, respectively. The content under severe drought was significantly lower than in the control and moderate drought. There was no significant difference in the content of soluble sugar in different treatments of *A. squarrosum* from HS on 11 August. Following a period of intense drought, the level of the substance reached its peak value (44.215 μg·g^−1^ FW) on 23 August, surpassing the control and moderate drought by 20.12% and 34.40%, respectively. Rehydration resulted in a reduction in soluble sugar of *A. squarrosum* from UD, and the content in control, moderate, and severe drought decreased by 14.05%, 30.44%, and 19.44% compared with before rehydration. With the exception of a rise during severe drought on 18 August, the soluble sugar content of *A. squarrosum* from HS was lower than before rehydration on 18 and 24 August.

On 11 August, there was no significant difference in soluble protein found in the two plants originating from different habitats. However, the content of *A. squarrosum* from UD increased when the stress level intensified on that particular day (Figure 3B). On 17 and 23 August, the content of *A. squarrosum* from UD in the control was 24.811 mg·g^−1^ FW and 28.340 mg·g^−1^ FW, respectively, which was significantly lower than under moderate and severe drought. Similarly, the content of *A. squarrosum* from HS under moderate drought was 22.750 mg·g^−1^ FW and 27.791 mg·g^−1^ FW, which was significantly lower than the control and severe drought. Rehydration led to an augmentation of the two species on 18 August and generated a decrease on 24 August. The content of *A. squarrosum* from UD exhibited a positive correlation with the level of stress intensity on 18 August. Under severe drought, its content reached the highest value (116.608 mg·g^−1^ FW). In comparison with the pre-rehydration levels, the content decreased by 1.64%, 22.33%, and 17.10% in the control, moderate, and severe drought conditions, respectively. The content of *A. squarrosum* from HS in the control and moderate drought conditions was 1.17 times and 1.32 times greater, respectively, compared with the content before rehydration. However, in severe drought conditions on 18 August, the content was 25.82% lower than before rehydration.

On 11 August, the proline content of *A. squarrosum* from UD and HS was 4.858 μg·g^−1^ FW and 5.385 μg·g^−1^ FW, respectively, which exceeded the levels observed during moderate and severe drought (Figure 3C). The proline content of *A. squarrosum* from various habitats exhibited an increase in response to escalating stress levels and prolonged duration on 17 and 23 August. On 23 August, the content of *A. squarrosum* from UD reached its highest value (11.720 μg·g^−1^ FW) under severe drought, which was 1.37 times and 1.22 times higher than the control and moderate drought. Similarly, the content of *A. squarrosum* from HS reached its highest level (9.215 μg·g^−1^ FW) under severe drought, which was 2.35 times higher than the concentration in the control group and 1.59 times higher than the concentration during moderate drought. On 18 August, the proline content of *A. squarrosum* from various habitats was greater in both control and moderate drought conditions compared with before rehydration However, in severe drought conditions, the proline content was lower. On 24 August, the content of *A. squarrosum* reduced due to rehydration, and this decline was further amplified as the level of drought stress increased.

### 3.4. Correlation Analysis

There was a substantial negative association between *F*v/*F*m and soluble sugar and SOD activity of *A. squarrosum* in different habitats (Figure 4). *A. squarrosum* from UD showed a strong positive correlation between CAT and soluble protein, chlorophyll, and MDA content. Additionally, there was a substantial positive correlation between proline concentration and soluble sugar content, and SOD activity. On the other hand, there was an inverse relationship between *F*v/*F*m and proline content. Regarding *A. squarrosum* from HS, we observed a substantial positive correlation between CAT activity and SOD activity, chlorophyll content, and MDA content. However, there was a notable negative correlation between CAT activity and *F*v/*F*m.

## 4. Discussion

Dry and semi-arid regions are frequently experiencing extreme weather events due to global climate change [31,32]. As a result, there is an increasing interest in researching the physiological and individual adaptation processes of plants in relation to changes in climate change. The researchers are to gain insight into the mechanisms by which desert and sandy plants adapt and develop in their harsh environments [33]. Chlorophyll is an important pigment in plant cells that plays a vital role in the process of photosynthesis. The level of chlorophyll in a plant has a direct impact on its capacity to carry out photosynthesis and its ability to withstand environmental influences [34,35]. The chlorophyll fluorescence parameter exhibits a robust correlation with plant photosynthesis and may accurately evaluate the real efficiency of plant photosynthesis under unfavorable circumstances. Additionally, it can be employed to measure the degree of stress that plants undergo as a result of external influences [36,37]. *F*v/*F*m represents the peak photochemical efficiency of PSII in the dark, demonstrating the optimal degree of light energy conversion efficiency in plants under intense light exposure [38,39]. Several studies have shown a significant reduction in chlorophyll levels with increasing levels of stress and longer durations of stress. The decrease in chlorophyll levels is accompanied by a decrease in chlorophyll production and an increase in its decomposition, leading to a reduced capacity to absorb and transfer light energy. The disruption of chloroplast structure also impedes photosynthesis and accelerates leaf wilting [40,41,42,43]. After performing our experiment, we observed that the leaves of both species in the control group displayed a phenomenon of turning yellow after being exposed to control watering for a period of 10 days. In addition, we observed a decrease in the chlorophyll levels of these leaves compared with those that were subjected to moderate and severe drought conditions. Throughout the experimental period, the concentration of *A. squarrosum* increased as the level of stress escalated, and the concentration after rehydration was lower than the concentration before rehydration. This implies that a plentiful supply of water facilitates the proliferation of *A. squarrosum*. Plant development and chlorophyll production can be affected by certain plants, growth phases, and levels of drought stress [11]. *A. squarrosum* is a pioneering plant species that is exceptionally proficient in stabilizing sand, which is the underlying reason for this phenomenon. The plant exhibits exceptional resistance to stress and drought, and it counteracts the adverse impacts of dryness by either increasing or sustaining a high concentration of photosynthetic pigments [44]. The chlorophyll content and *F*v/*F*m of *A. squarrosum* from the UD site demonstrated higher values in comparison with those from the HS site, indicating that the species from the UD site exhibited a strong ability to endure dry conditions. Plants often thrive in dry regions, which helps them develop the ability to withstand drought. *Agriophyllum squarrosum* can improve its capacity to endure dry circumstances by maintaining the stability of its photosynthetic pigments. The process of long-term adaptation to various habitats has resulted in the emergence of unique mechanisms of adaptation within the same species [45].

Plants produce reactive oxygen species (ROS) in response to environmental stress as a reaction to external pressure. Even a small amount of ROS can operate as a signaling molecule, initiating the activation of plant defense mechanisms. An overabundance of ROS leads to the oxidation of lipids in cell membranes, causing a significant accumulation of malondialdehyde. The membrane’s content level indicates the level of lipid peroxidation, the extent of membrane damage, and the plants’ resilience to adversity [46,47]. Previous studies have shown that the level of malondialdehyde increases with the intensity and duration of stress [48,49]. However, our research uncovered that the content of both species was diminished during periods of moderate and severe drought in comparison with the control group. This provides additional evidence that an excessive amount of moisture has a harmful impact on *A. squarrosum*. After being rehydrated, the level of malondialdehyde decreased under both moderate and severe drought conditions. In addition, the *F*v/*F*m ratio, with the exception of the rehydration after 13 days of moderate drought in *A. squarrosum* from the UD site, reverted to the control level during the later stage of rehydration, demonstrating an overcompensation effect. The findings indicated that both plants had a reduction in the suppression of the photoreaction center when they were provided with water again during periods of moderate and severe drought. In addition, the capability of photosynthetic electron transport returned to its usual level.

In plant cells, the reactive oxygen species generated during aerobic metabolism and their elimination by antioxidant defense systems exist in a dynamic balance of redox. However, if the amount of external environmental stress exceeds the plant’s antioxidant defense system’s capacity to eliminate it, there will be a substantial accumulation of ROS, which would interfere with the plant’s regular metabolic processes [50]. Antioxidant enzymes are of the utmost importance in plants when confronted with unfavorable circumstances. They possess the capacity to reduce or diminish the damage caused by ROS and effectively counteract the harmful consequences of ROS during stressful situations. As a result, they are closely linked to the capacity to endure stress [51]. After a period of drought and subsequent rehydration, the activity levels of POD, SOD, and CAT were increased in both species. The findings indicated that the three enzymes inhibited the occurrence of membrane lipid peroxidation, hence reducing membrane harm in the context of climate change. This discovery offers additional proof of *A. squarrosum’*s capacity to acclimate to more demanding environmental conditions and establish itself as a species that leads the way. The antioxidant enzyme activities of *A. squarrosum* from different places displayed diverse patterns as the period of drought increased. Nevertheless, the activity of *A. squarrosum* from UD surpassed that of HS, suggesting the presence of a long-term physiological adaption mechanism that has evolved in arid conditions. After the rehydration process, the POD and SOD activities of *A. squarrosum* from the UD site increased, whereas the CAT activity decreased. This study revealed that an excessive amount of water has an adverse effect on the organism’s normal physiological metabolism. Furthermore, it was demonstrated that the enzymes POD and SOD played a crucial role in alleviating the damage produced by excessive water. After a 7-day period of water deprivation, the levels of these enzymes in *A. squarrosum* from HS were seen to be drastically diminished upon rehydration. After being deprived of water for 13 days, the levels of three antioxidant enzymes showed a considerable rise when water was reintroduced, in response to the severe drought circumstances. These enzymes worked together to alleviate the harmful consequences of extended drought. 

Studies have shown that osmoregulation plays a crucial role in responding to stress [52]. Plants use physiological metabolic processes to produce osmoregulatory chemicals, which helps lower osmotic potential and maintain proper cell turgor pressure. Plants are able to adjust to harsh circumstances, perform regular physiological functions, and improve their resistance and adaptability [53]. There was no significant difference in the levels of soluble protein after the water control. Nevertheless, the levels of soluble sugar and proline in both species were elevated in the control group in comparison with the groups exposed to moderate and severe drought. The observed pattern of malondialdehyde levels agrees with the hypothesis that the xerophyte *A. squarrosum* is more suitably adapted to drought circumstances. In addition, the concentrations of three osmoregulators exhibited a rise in both species when subjected to moderate and severe drought conditions, in conjunction with the duration of the drought. This implies that, when drought stress becomes more severe, many osmotic regulators are engaged in maintaining osmotic equilibrium. Following a 13-day period of water deprivation, the reintroduction of water did not result in any discernible alteration in the proline levels of *A. squarrosum* from the HS site during a mild drought. Nevertheless, the concentrations of three osmotic regulators in both species were reduced compared with the levels before the restoration of water. Proline levels in *A. squarrosum* were comparatively low in different habitats throughout both drought and rehydration periods. This implies that *A. squarrosum* predominantly regulates cell turgor pressure by accumulating soluble sugar and soluble protein in order to adapt to adverse situations resulting from variations in water supply. Proline, however, has a subordinate role in osmotic control. The concentrations of soluble sugar and soluble protein in *A. squarrosum* from the UD site were higher than those from the HS site, which contributed to its improved ability to withstand drought.

## 5. Conclusions

Our study clearly demonstrated that supplying a sufficient quantity of water (irrigating up to 60–65% of field capacity) surpassed the required amount and had a detrimental impact on xerophyte. *A. squarrosum* showed improved resistance to drought and underwent significant physiological and metabolic changes during an extended period of adaptation and evolution in many habitats. Nevertheless, plants from UD exhibited greater drought resilience compared with the plants from HS. The coordinated action of enzymes (e.g., POD, SOD, and CAT) minimized membrane damage during drought circumstances. In addition, the plant regulated osmotic pressure and water balance by rapidly accumulating soluble sugar and soluble protein. A water deficiency resulted in varying levels of damage to the plants, and neither entirely recovered when exposed to rehydration throughout the indicated timeframe. These findings improve the comprehension of plant resistance theory and offer a theoretical basis for future ecological engineering and the construction of artificial vegetation.

## Figures and Tables

**Figure 1 plants-13-02601-f001:**
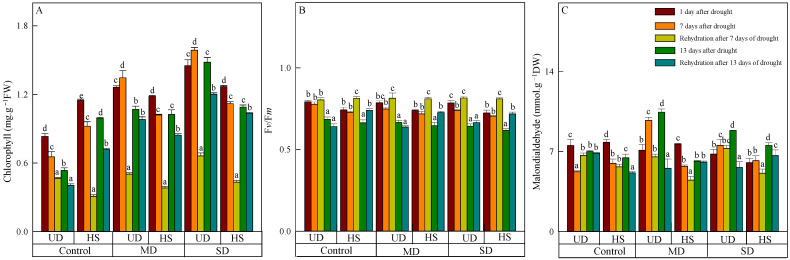
Changes in chlorophyll (**A**), quantum efficiency of photosystem II (*F*v/*F*m) (**B**), and malondialdehyde (**C**) of *A. squarrosum* from Urat desert steppe (UD) and Horqin sandy land (HS) under drought and rehydration. Using the ANOVA procedure, we compared the levels of chlorophyll, *F*v/*F*m, and malondialdehyde, and subsequent LSD Fisher’s test was employed. Bars labeled with different lowercase letters represent significant differences among days for a given treatment (*p* < 0.05). MD denotes moderate drought (with the field capacity of 40–45%); SD denotes severe drought (with the field capacity of 20–25%).

**Figure 2 plants-13-02601-f002:**
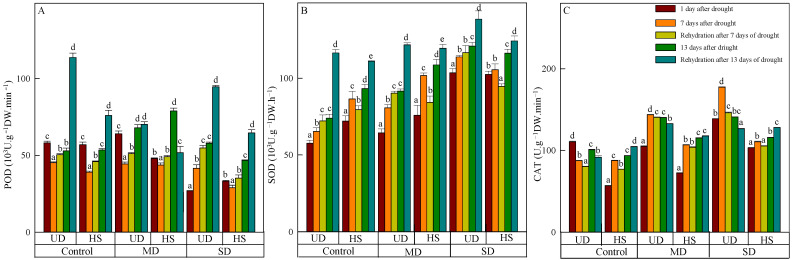
Changes in peroxidase (POD) (**A**), superoxide dismutase (SOD) (**B**), and catalase (CAT) (**C**) of *A. squarrosum* from Urat desert steppe (UD) and Horqin sandy land (HS) under drought and rehydration. Using the ANOVA procedure, we compared the levels of POD, SOD, and CAT, and subsequent LSD Fisher’s test was employed. Bars labeled with different lowercase letters represent significant differences among days for a given treatment (*p* < 0.05). MD denotes moderate drought (with the field capacity of 40–45%); SD denotes severe drought (with the field capacity of 20–25%).

**Figure 3 plants-13-02601-f003:**
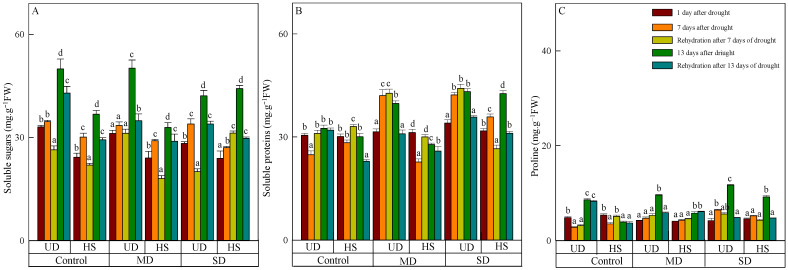
Changes in soluble sugars (**A**), soluble proteins (**B**), and proline (**C**) of *A. squarrosum* from Urat desert steppe (UD) and Horqin sandy land (HS) under drought and rehydration. Using the ANOVA procedure, we compared the levels of soluble sugars, soluble proteins, and proline, and subsequent LSD Fisher’s test was employed. Bars labeled with different lowercase letters represent significant differences among days for a given treatment (*p* < 0.05). MD denotes moderate drought (with the field capacity of 40–45%); SD denotes severe drought (with the field capacity of 20–25%).

**Figure 4 plants-13-02601-f004:**
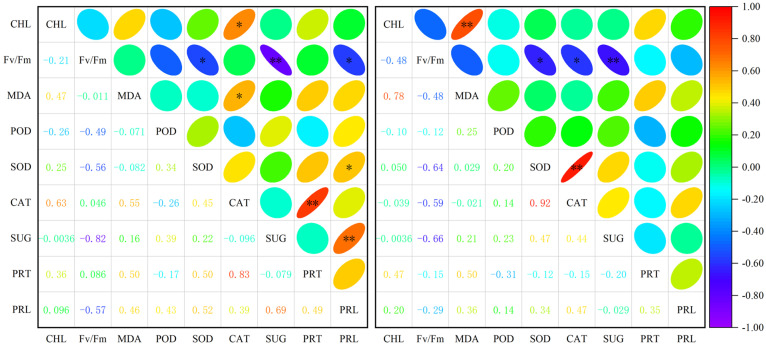
Correlation analysis among physiological indices of *A. squarrosum* from Urat desert steppe (**Left**) and Horqin sandy land (**Right**). * *p* ≤ 0.05, ** *p* ≤ 0.01. CHL denotes chlorophyll; MDA denotes malondialdehyde; SUG denotes soluble sugar; PRT denotes soluble protein; PRL denotes proline.

**Table 1 plants-13-02601-t001:** Difference in growth characteristics of *A. squarrosum* from Urat desert steppe (UD) and Horqin sandy land (HS) under natural drought. Bars labeled with different lowercase letters represent significant differences among drought treatments on a given day. MD denotes moderate drought (with the field capacity of 40–45%); SD denotes severe drought (with the field capacity of 20–25%).

Item	Height (cm)		Weight (g)		Color	
	UD	HS	UD	HS	UD	HS
Control	28.80 ± 2.24 a	32.70 ± 2.36 a	7.18 ± 0.56 a	8.85 ± 0.29 a	Chartreuse	Chartreuse
MD	29.10 ± 1.36 a	32.30 ± 1.09 a	7.66 ± 1.34 ab	8.34 ± 0.33 a	Chartreuse	Green
SD	29.320 ± 0.98 a	31.90 ± 1.20 a	8.15 ± 0.35 b	7.63 ± 0.78 b	Green	Green

## Data Availability

The original contributions presented in the study are included in the article. Further inquiries can be directed to the corresponding author.

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
