# Peer review of "Effect of Drought and Rehydration on Physiological Characteristics of Agriophyllum squarrosum (L.) Moq. in Different Habitats"

_plants, 2024, doi:10.3390/plants13182601_

Round 1

Reviewer 1 Report

Comments and Suggestions for Authors

Dear Authors,

The general idea of the manuscript “Effect of drought and rehydration on physiological characteristics of Agriophyllum squarrosum in different habitats” is interesting and corresponds to the field of the journal. However, the article contains a number of significant shortcomings

  1. Professional and language editing required
  2. Divide chapter 2.3 into appropriate parts and describe the analyzes in more detail: extraction method, reagents and kits, instruments, references to the literature used
  3. The lack of morphometric data after stress (weight, height, color) is very confusing and makes the entire presented data set meaningless
  4. If the authors compare 2 variants of plants from two habitats, then it is completely unclear why all the data are presented on different histograms? I strongly recommend that the histograms be redone so that the indicators of these two plant options can be compared with each other.
  5. Perhaps by solving these problems, the meaning of the work will become more clear to readers. The introduction and discussion do not fully illuminate the meaning of the data obtained

Best regards

Author Response

Dear reviewer:

Thank you for your comments concerning our manuscript entitled “Effect of drought and rehydration on physiological characteristics of Agriophyllum squarrosum in different habitats”. Those comments are very helpful for revising and improving our paper, as well as the important guiding significance to our researches. We have studied comments carefully and made correction. The main corrections in the paper and the responds to the reviewers’ comments are as follows.

Dear Authors,

The general idea of the manuscript “Effect of drought and rehydration on physiological characteristics of Agriophyllum squarrosum in different habitats” is interesting and corresponds to the field of the journal. However, the article contains a number of significant shortcomings

  1. Professional and language editing required.

According to the reviewer’s suggestion, we have edited the manuscript by a native English speaker.

  1. Divide chapter 2.3 into appropriate parts and describe the analyzes in more detail: extraction method, reagents and kits, instruments, references to the literature used.

We have added much more detailed description on chapter 2.3.

Blades fixed with liquid nitrogen were extracted using a chilled buffer (2% polyvinylpolypyrrolidone, 50 Mm phosphate) and centrifuged the extract at 15000 g for 20 minutes, and the supernatant was used for the determination of various physiological indicators [1]. The malondialdehyde (MDA) content was determined spectrophotometrically at 450 nm, 532 nm and 600 nm using the thiobarbituric acid method [28]. The activities of antioxidant enzymes (POD, SOD, CAT) were specifically measured according to An et al [29].POD activity was determined using guaiacol method, with 1 μmol guaiacol oxidation per gram of plant fresh weight per minute as an enzyme activity unit. SOD activity was determined using nitrogen blue tetrazole (NBT) photoreduction method, with 50% inhibition of NBT photoreduction per gram of plant fresh weight as one enzyme activity unit. CAT activity was determined using hydrogen peroxidation-iodimetry, and 1 μmol H2O2 was decomposed per gram of fresh plant weight per minute as an enzyme activity unit [11]. The contents of osmoregulator were definitely measured according to Luo et al [30]. The soluble sugars content was measured with anthrone reaction. The soluble proteins content was determined with coomassie brilliant blue method and the proline content was determined with ninhydrin method [12]. The activities of antioxidant enzymes and the contents of osmoregulatory substances and MDA were measured using a spectrophotometer (Shimadzu Corporation, Japan).

  1. The lack of morphometric data after stress (weight, height, color) is very confusing and makes the entire presented data set meaningless.

As reviewer suggested that we have added much more detailed description on morphometric characteristics after stress.

Table 1. Difference on growth characteristics of A. squarrosum from Urat desert steppe (UD) and Horqin sandy land (HS) under natural drought. Bars labeled with different lowercase letters represent significant differences among drought treatments on a given day. MD denotes moderate drought (with the field capacity of 40–45 %); SD denotes severe drought (with the field capacity of 20–25 %).

Item

Height (cm)

Weight (g)

Color

UD

HS

UD

HS

UD

HS

Control

28.80±2.24a

32.70±2.36a

7.18±0.56a

8.85±0.29a

Chartreuse

Green

MD

29.10±1.36a

32.30±1.09a

7.66±1.34ab

8.34±0.33a

Chartreuse

Green

SD

29.320±0.98a

31.90±1.20a

8.15±0.35b

7.63±0.78b

Green

Green

  1. If the authors compare 2 variants of plants from two habitats, then it is completely unclear why all the data are presented on different histograms? I strongly recommend that the histograms be redone so that the indicators of these two plant options can be compared with each other.

We have corrected figures according to the reviewer’s comments.

  1. Perhaps by solving these problems, the meaning of the work will become more clear to readers. The introduction and discussion do not fully illuminate the meaning of the data obtained.

According to the reviewer’s suggestion, we have revised the introduction and discussion.

We tried our best to improve the manuscript and made some changes in the manuscript. These changes will not influence the content and framework of the paper. And here we did not list the changes but marked in red in revised paper.

Once again, thank you very much for your comments and suggestions.

Reviewer 2 Report

Comments and Suggestions for Authors

The manuscript is well-written and well-structured. The results are well presented. The selection of literature is appropriate. However, I believe that some revisions and improvements could strengthen the manuscript

•            The full names of the enzymes should be included in the abstract, rather than just their abbreviations.

·                In the Methods and Materials section, explaining why those two specified locations were chosen for plant seed sampling is necessary.

·                In lines 66,93,350,366 etc. it is necessary to separate the words, it is necessary to review the entire text

·                In Figures 1 and 2, it is necessary to specify the meaning of MD and SD

·                The discussion should be improved. It does not effectively explain or compare the obtained results with previous studies properly. It is necessary to compare the obtained results with the location from which the seeds of the tested plants originated

·                Photographs from the experiment should be included

Author Response

Dear reviewer:

Thank you for your comments concerning our manuscript entitled “Effect of drought and rehydration on physiological characteristics of Agriophyllum squarrosum in different habitats”. Those comments are very helpful for revising and improving our paper, as well as the important guiding significance to our researches. We have studied comments carefully and made correction. The main corrections in the paper and the responds to the reviewers’ comments are as follows.

The manuscript is well-written and well-structured. The results are well presented. The selection of literature is appropriate. However, I believe that some revisions and improvements could strengthen the manuscript.

  1. The full names of the enzymes should be included in the abstract, rather than just their abbreviations.

We have added full names of the enzymes in the abstract.

peroxidase (POD), superoxide dismutase (SOD), catalase (CAT).

  1. In the Methods and Materials section, explaining why those two specified locations were chosen for plant seed sampling is necessary.

According to the reviewer’s suggestion, we have added more detailed description on Methods and Materials section.

Land degradation and water constraint have made vegetation restoration more difficult in both areas during the past two decades. Under the background of global warming, frequent extreme precipitation events would dramatically influence the community structure and function, individual germination characteristics and physiological response in UD and HS ecosystem.

  1. In lines 66, 93, 350, 366 etc. it is necessary to separate the words, it is necessary to review the entire text

We have separated the words according to the reviewer’s comments.

50-400 mm; 1935 mm; 15-25 ℃; 663 and 645 nm; 9.722 mmol.g-1FW and 10.453 mmol.g-1FW; 7.521 mmol.g-1FW; 113.768 H2O2.g-1FW.min-1; 120.838 μg.FW-1.hr-1; 49.968 μg.g-1FW, 50.162 μg.g-1FW and 42.133 μg.g-1FW; 44.215 μg.g-1FW; 24.811 mg.g-1FW; 28.340 mg.g-1FW; 22.750 mg.g-1FW; 27.791 mg.g-1FW; 116.608 mg.g-1FW; 4.858 μg.g-1FW and 5.385 μg.g-1FW; 11.720 μg.g-1FW; 9.215 μg.g-1FW; When re-watering after 13 days of drought; When re-watering after 13 days of drought.

  1. In Figures 1 and 2, it is necessary to specify the meaning of MD and SD

As reviewer suggested that we have added much more detail on Figures.

  1. The discussion should be improved. It does not effectively explain or compare the obtained results with previous studies properly. It is necessary to compare the obtained results with the location from which the seeds of the tested plants originated

According to the reviewer’s suggestion, we have revised the discussion.

  1. Photographs from the experiment should be included.

Thank you very much for your recognition of this study. Your suggestions can make our research more comprehensive. We planted a total of 6 plants during our experiment, and all the photos we took were group photos, and we didn't take them separately for A. squarrosum. We are sorry for our negligence in not having suitable photographs for the paper.

We tried our best to improve the manuscript and made some changes in the manuscript. These changes will not influence the content and framework of the paper. And here we did not list the changes but marked in red in revised paper.

Once again, thank you very much for your comments and suggestions.

Round 2

Reviewer 1 Report

Comments and Suggestions for Authors

the work is approaching publication. However, there are a few small notes:

1. move table 1 to the appropriate results chapter.

2. indicate the full name of the plant in the title, abstract and when first mentioned in the introduction (Agriophyllum squarrosum (L.) Moq.).

3. Indicate the method of statistical processing in the figure legends and the P value (P>0.05?).

4. Edit the keywords: add antioxidant activity and, in my opinion, Urat desert steppe; Horqin sandy land is better replaced with desert steppe and sandy land

Author Response

Dear reviewer:

Thank you for your comments concerning our manuscript. We have studied comments carefully and made correction. The main corrections in the paper and the responds to the reviewers’ comments are as follows.

  1. move table 1 to the appropriate results chapter.

We have moved table 1to the results chapter.

  1. indicate the full name of the plant in the title, abstract and when first mentioned in the introduction (Agriophyllum squarrosum (L.) Moq.).

We have corrected plant name according to the reviewer’s comments.

  1. Indicate the method of statistical processing in the figure legends and the P value (P>0.05?).

As reviewer suggested that we have added statistical processing and P value on figure legends.

  1. Edit the keywords: add antioxidant activity and, in my opinion, Urat desert steppe; Horqin sandy land is better replaced with desert steppe and sandy land.

According to the reviewer’s suggestion, we have revised the keywords.

Once again, thank you very much for your comments and suggestions.
